# Application of Synthetic Biology Approaches to High-Yield Production of Mycosporine-like Amino Acids

**Varsha K. Singh** [1], **Sapana Jha** [1], **Palak Rana** [1], **Amit Gupta** [1], **Ashish P. Singh** [1], **Neha Kumari** [1], **Sonal Mishra** [1], **Prashant R. Singh** [1], **Jyoti Jaiswal** [1] and **Rajeshwar P. Sinha** [1,2,*]

[1] Laboratory of Photobiology and Molecular Microbiology, Centre of Advanced Study in Botany, Institute of Science, Banaras Hindu University, Varanasi 221005, India; kumarivarsh931@gmail.com (V.K.S.); jha422607@gmail.com (S.J.); ranapalak271@gmail.com (P.R.); amitgupta.bhu15@gmail.com (A.G.); singhashishpratap24@gmail.com (A.P.S.); nehayadavbhu123@gmail.com (N.K.); mishrasona227@gmail.com (S.M.); prasinghbhu@gmail.com (P.R.S.); jyotij273@gmail.com (J.J.)

[2] University Center for Research & Development (UCRD), Chandigarh University, Chandigarh 140413, India

[*] Correspondence: rpsinhabhu@gmail.com; Tel.: +91-542-2307147; Fax: +91-542-2366402

**Abstract:** Ultraviolet (UV) radiation reaching the Earth's surface is a major societal concern, and therefore, there is a significant consumer demand for cosmetics formulated to mitigate the harmful effects of UV radiation. Synthetic sunscreens being formulated to block UV penetration include inorganic metal oxide particles and organic filters. Lately, organic UV-absorbing compounds are manufactured from non-renewable petrochemicals and, as a result, there is a need to develop a sustainable manufacturing process for efficient, high-level production of a naturally occurring group of UV-absorbing compounds, namely mycosporine-like amino acids (MAAs), for use as a sunscreen additive to skincare products. Currently, the commercial production of MAAs for use in sunscreens is not a viable proposition due to the low yield and the lack of fermentation technology associated with native MAA-producing organisms. This review summarizes the biochemical properties of MAAs, the biosynthetic gene clusters and transcriptional regulations, the associated carbon-flux-driving processes, and the host selection and biosynthetic strategies, with the aim to expand our understanding on engineering suitable cyanobacteria for cost-effective production of natural sunscreens in future practices.

**Keywords:** biosynthetic gene clusters; mycosporine-like amino acids; sunscreens; synthetic biology



## 1. Introduction

Chemicals, including oxybenzone, ZnO, and $TiO_2$, are frequently used in skincare products to protect against skin damage from UV rays. However, these chemicals have several negative effects on human health and the environment. Several common chemical UV sunscreen filters are absorbed by the skin and enter the bloodstream. The usage of chemical and inorganic sunscreens has increased in recent decades to counteract the harmful effects of UV radiation, but this practice has been linked to a number of skin-related diseases, including skin allergies, rashes, premature aging, dermal cancer, and other skin problems [1]. As a result, there is an increasing need to find bio-based sunscreen chemicals that are efficient, safe, sustainable, and that have the ability to prevent UV-induced damage and boost the effectiveness of natural sunscreens [2]. Mycosporines and mycosporine-like amino acids (MAAs), synthesized by both prokaryotic as well as eukaryotic organisms such as fungi, cyanobacteria, and algae, are natural UV protectants [3]. Mycosporines, which are primarily produced in fungi, consist of the nitrogen substituent of an amino acid or an imino alcohol at the C3 position, forming the cyclohexenone ring, while MAAs have an additional nitrogen substituent conjugated via imine linkage, forming the cyclohexenimine core structure [4]. MAAs are low molecular weight (<400 Da), water-soluble, and colorless UV protectants. They have a high molar extinction coefficient ($\varepsilon = 28{,}100\text{–}50{,}000\ M^{-1}\ cm^{-1}$),

and their maximum absorption wavelength lies within 309 to 362 nm. These compounds possess a wide array of bioactivities, such as antioxidative, anti-inflammatory, anti-aging, and antitumor activities [1]. Majority of research on cyanobacterial MAAs focuses on specific areas, such as the identification of bioactive compounds, in-depth examination of their molecular mechanisms of action, and the evaluation of their bioactivities via in vitro, in silico, and in vivo analyses.

Lately, the gene-centric approach, or bottom-up approach, has been used to explain the biosynthetic abilities of cyanobacteria by combining in silico studies with functional genomics to link the genomic context, known as biosynthetic gene clusters (BGCs), to target desired metabolites [5]. These days, it is easier to discover the cyanobacterial BGCs of diverse compound families mainly through a number of in silico analysis tools available for cyanobacterial genomic data, including antiSMASH and PRISM [6–8]. These resources have made it easier to understand the cyanobacterial MAAs biosynthesis.

Most of the identified cyanobacterial BGCs are silent, making them a greater challenge in MAAs research. Synthetic biology approaches, such as metabolic engineering and strain mutagenesis, have been employed to activate the silent BGCs of other bacterial phyla, for instance, actinomycetes. However, the use of these approaches is less reported for the discovery of cyanobacterial products, mainly because of the slow growth of cyanobacterial strains and because these are genetically less amenable, suggesting an open area to discover and characterize the potential novel compounds from cyanobacteria [9]. The production of the encoded molecule on a large scale for usage at an industrial level has been successfully accomplished through the introduction of targeted metabolite biosynthetic genes into a suitable heterologous host. In order to produce specific metabolite analogues or to maximize the production yield, genetic contents can be redesigned with the help of heterologous expression of the targeted secondary metabolite [10,11]. In this paper, we present recent advances in the production of cyanobacterial MAAs in suitable heterologous hosts. We will go into detail on a number of metabolic engineering and synthetic biology techniques for constructing BGCs and enhancing transcriptional and translational productivity.

## 2. Application of MAAs (Sunscreen) in Cosmetics

The idea of sunscreen usage has changed from being seen as only a UV-protective compound to provide broad-spectrum defense against photoaging, DNA damage, and dyspigmentation [12]. When selecting a sunscreen, one should take into account the possibility that infrared and visible light contribute to photoaging. UV rays and visible light will be shielded against by using a broad-spectrum tinted sunscreen with SPF of at least 30, which will lessen their impacts on photoaging [12]. For mending skin aging, healing, and avoiding wrinkle development, several microalgal extracts, including those from *Spirulina platensis*, *Chlorella vulgaris*, and *Dunaliella salina*, can be utilized [13,14]. The extracts might provide creative and potential replacements for current cosmetics, and they encourage the development of new uses for cosmetics.

### 2.1. Photoprotection Prospects of MAAs

MAAs, which are hydrophilic and colorless, are synthesized by marine cyanobacteria [15,16], microalgae, macroalgae, etc., that function as an antioxidant by reducing ROS-induced DNA damage and as a photoprotectant by protecting cells from UVR [16,17]. Only a small proportion of these so-called "broad-spectrum sunscreens" are truly efficient at blocking both UV-A and UV-B rays [18]. The strong ability of MAAs to absorb UVR between 309 and 362 nm defines them as a broad-spectrum sunscreen, making it essential to incorporate MAAs as a UV-filter agent in sunscreens [19]. Due to their high photostability, and resistance to heat, pH changes, and various solvents, they can be a very stable cosmetic product [20]. The first sunscreen, called Helioguard 365, was developed by the Swiss Firm Mibelle AG Biotechnology, utilizing a naturally occurring UV-blocking component known as MAA, that contains a certain amount of porphyra-334 and shinorine derived from red algae, *Porphyra umbilicalis* [21,22]. The MAAs derived from the algae *Dunaliella*, *Arthrospira*,

and *Chlorella* function as sunscreens to mitigate the harm caused by UVR. *Odontella aurita*, a kind of microalgae, also showed potential free radical scavenging action. Coelastrin A and Coelastrin B, two new MAAs from *Coelastrella rubescens*, have photoprotective characteristics [23]. The recently discovered MAAs from *Klebsormidium*, klebsormidin A and klebsormidin B, demonstrated that UVR exposure dramatically induces their production and intracellular enrichment, indicating the role of these molecules as natural UV sunscreens [24]. However, there are still very few MAAs that are commercially available [25]. Additionally, there are two commercial products called Helioguard®365 and Helionori® that both include an extract with an enhanced or specified MAAs content [26].

Porphyra-334 and shinorine both provided concentration-dependent protection when UV-A exposure was evaluated. The recommended dosage for the best protection was 5 μg mL$^{-1}$ [27]. According to Schmid et al. [26], the formulation they utilized for their study contained 5% of Helioguard®365 (final MAA concentration of 0.005%), the same base with 4% of a synthetic UV-B sunscreen and 1% of a synthetic UV-A sunscreen. It is well-known that Helioguard®365 has anti-aging and photoprotective properties. In a dose-dependent manner, Helioguard®365 concentrations of 0.125% and 0.25% increased cell viability; at 0.25% of Helioguard®365, cell viability was increased by as much as 97.8% [26]. On applying a 2% concentration of Helioguard®365 to the cell lines, the SPF value of the sunscreen increases from 7.2 to 8.2. Sunscreen containing porphyra-334 with shinorine and mycosporine-serinol in the ratio of 4.1:2.9% have the SPF value of 8.37 ± 2.12, whereas for porphyra-334 with shinorine and mycosporine-serinol when applied separately, the SPF value was observed from 4 to 6 [28]. An algal extract having a combination of palythine, asterina-330, shinorine, porphyra-334, and palythinol obtained from *H. cornea* and *G. longissima* showed a concentration-dependent increase in the SPF value. At a density of 13.9 mg DW of algae per cm$^{-2}$, the SPF values of *G. longissima* and *H. cornea*, respectively, were found to be 7.5 and 4.8. Both algal extracts increased TNF-α and IL-6 production, while exhibiting no cytotoxicity toward human cells [29]. Shinorine and porphyra-334 extract, which is found in liposomes and is encapsulated in Helioguard®365, reduce the lipid peroxidation of human skin, improve skin firmness and smoothness, and play a role in the prevention of premature aging. Helionori® can prevent sunburn and preserve membrane lipids because it has mycosporine-palythine, porphyra-334, and shinorine, which make it photo- and heat-stable [30]. Biotechnologically, MAAs can be used for a variety of commercial purposes, such as in dietary supplements, medicine, functional organic devices, and others. Therefore, the commercialization of MAAs with multiple uses is an exciting prospective endeavor [31].

## 2.2. Anti-Aging Prospects of MAAs

It might be challenging to characterize skin aging since it is a result of systems involving heredity and environmental variables [32]. Unlike chronological aging, premature skin aging, or photoaging, is brought on by exposure to environmental stress [33]. Clinical manifestations of photoaging include dryness, hyper- and hypo-pigmentation, leathery texture, and wrinkles [33]. As already mentioned, cyanobacteria synthesize molecules that may be used in skin moisturization, UV protection (MAAs and SCY), and shielding against ROS (MAAs, SCY, PBPs, and polyphenols), making them intriguing for use in skin anti-aging treatments [16]. These effects were also mentioned for fibroblasts, which are responsible for the skin's firmness and elasticity, as well. The extracellular matrix (ECM), which mainly consists of collagen and elastin and provides firmness and elasticity, is produced by fibroblasts within the dermis [34]. In studies employing normal human dermal fibroblasts (nHDFs) exposed to UV-B radiation, it was shown that extracts from *Arthrospira platensis* increased cell viability and reduced DNA damage by inhibiting thymine dimers and matrix metalloproteinases (MMP) [35]. Mycosporine-glycine, porphyra-334, palythine, and shinorine, the most prevalent MAAs, were examined for their UV-protective properties in recent research using a variety of cell models (human skin fibroblasts and HaCaT keratinocytes) to demonstrate their effectiveness as possible sunscreens [20]. Additionally, a number of anti-aging

actions, particularly those that target the elastic fibers of the extracellular matrix (ECM), such as collagen and elastin, as well as their remodeling enzymes, have been identified [36,37]. Furthermore, there are two commercial products called Helioguard®365 and Helionori® that both include an extract with an enhanced or specified MAAs content [26].

The prevention of skin aging is a result of several mechanisms, including skin hydration, UV protection, stimulation of fibroblast growth, and an increase in antioxidant capacity. Cyanobacteria produce substances that have been shown to interfere with all these processes in the aforementioned areas, making them attractive for cosmetics meant to delay the onset of skin aging.

### 3. Biosynthesis and Genetic Regulation of MAAs

Despite having different molecular structures, mycosporines are composed of a typical cyclohexenone ring structure, which provides them the same spectral characteristics and absorption maxima [38]. Studies have demonstrated that a cyanobacterium or a cyanobacterial ancestor acted as the progenitor of the enzymes for MAAs production [39–43]. MAAs are synthesized in cyanobacteria using a four-enzyme pathway. It was discovered that different cyanobacterial species exhibit significant genetic variation in the *mys* gene cluster, which becomes involved in MAAs biosynthesis [44]. The latest studies on the biochemical processes and genetic research have contributed to developing knowledge of the fundamental steps that are involved in the biosynthesis of MAAs. *Anabaena variabilis* has been used to explain the initial step in the biosynthesis of MAAs. Therefore, it is believed that cyanobacteria were the first producers of MAAs, and the genes for MAA biosynthesis most likely spread to other organisms. The shikimate pathway is suggested as a potential biosynthetic route for MAAs [45]. MAAs are synthesized from phosphoenolpyruvate (PEP) and erythrose 4-phosphate (E4P) (an intermediate in the pentose phosphate pathway). PEP and E4P react to form 3-deoxy-D-arabinoheptulosonate-7-phosphate (DAHP), which is catalyzed by the enzyme DAHP synthase. DAHP is further involved in the formation of 3-dehydoquinate (3-DHQ), which is a six-membered carbon ring-like structure (Figure 1). The 3-DHQ produces 4-deoxygadusol (4-DG), which is subsequently followed by gadusol [42]. Through a separate route, sedoheptulose 7-phosphate (S7P), another intermediate product of the PPP, is converted into 4-DG in the presence of the enzyme dimethyl 4-degadusol synthase (DDGS; MysA) and an O-methyltransferase (O-MT; MysB) [46].

The common precursor for all MAAs in both pathways is 4-DG, and the conjugation of this precursor with a glycine molecule results in a simple mono-substituted cyclohexenone-type MAA, called mycosporine-glycine (MG). This further acts as a common intermediate in the production of di-substituted (aminocyclohexene imine-type) MAAs, such as porphyra-334 (P-334) and shinorine (SH). In some species, MAAs cannot be synthesized simply by substituting an amino acid for 4-DG, but they are produced through shifting the side chains of amino acids via condensation (for the esterification process and amidation), dehydration (for the formation of double bonds), decarboxylation (for chain shortening), and oxidation and reduction (for hydroxylation) [45]. The shikimic acid pathway has been scientifically investigated in order to further comprehend how it affects the production of 4-DG, which is the precursor molecule for MAAs biosynthesis [15]. A shikimate intermediate, [Ue$^{14}$C] 3-dehydroquinic acid, was selectively taken up by *Trichothecium roseum,* which resulted in structurally similar fungal mycosporines' production [47]. When [$^{14}$C] pyruvate was studied for MAAs biosynthesis via phosphoenolpyruvate, radiolabeled MAAs having more specificity were formed, while [$^{14}$C] acetate (polyketide pathway) was unable to produce MAAs [48]. Each species requires a different set of biosynthetic pathways for producing MAAs, and these pathways are affected by many abiotic factors, such as salinity, temperature, humidity, moisture loss, and other abiotic factors [49]. Cyanobacteria were not able to produce MAAs in the presence of tyrosine, which inhibits the shikimic acid pathway. A similar result was found in the case of *Nostoc commune* when it was exposed to glyphosate, which inhibits the shikimic acid pathway. Therefore, the shikimic acid pathway can be suppressed or inhibited by exogenous tyrosine or glyphosate [42].

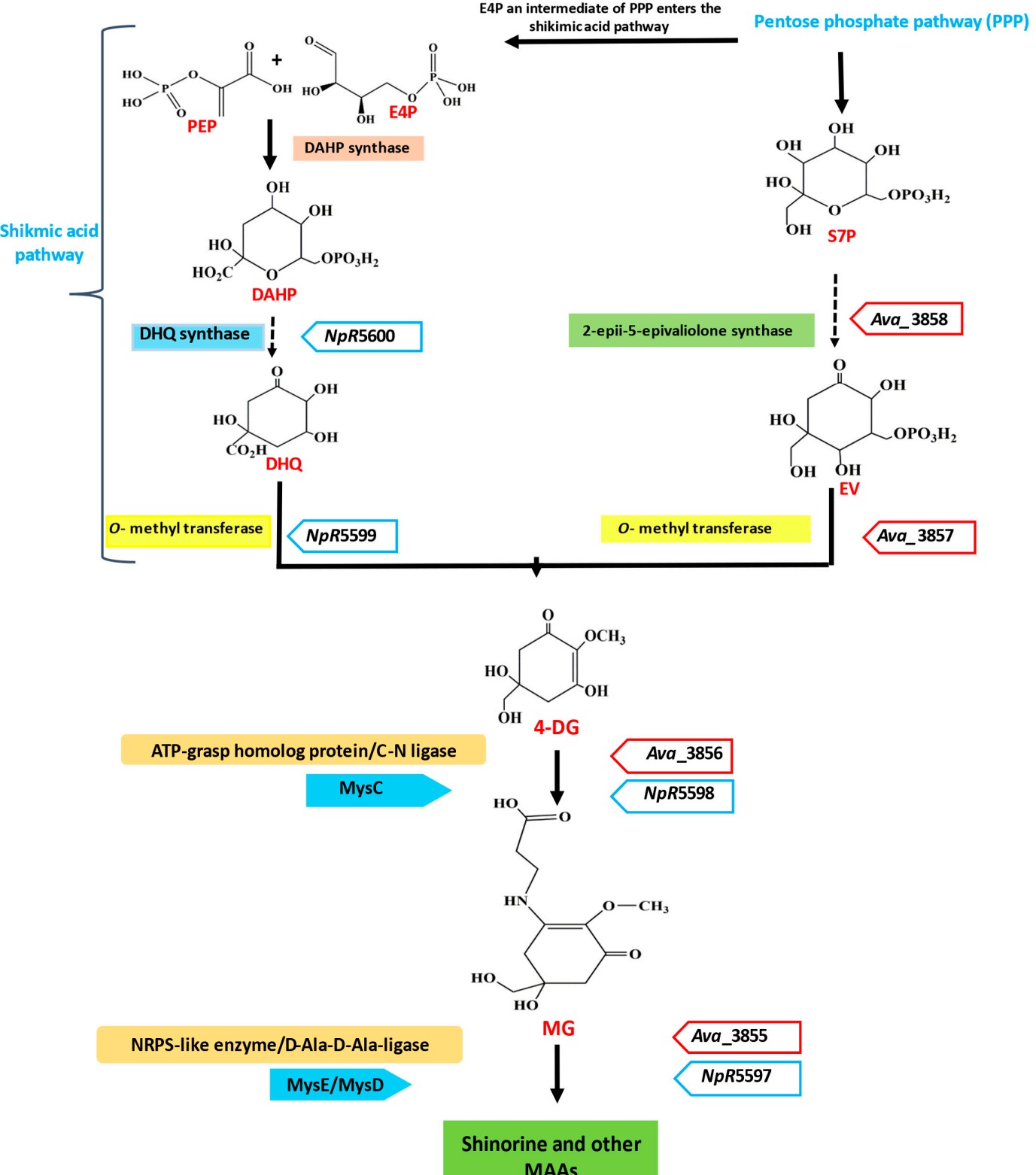

**Figure 1.** Schematic representation of the biosynthetic pathway of MAAs in cyanobacteria. Synthesis from the shikimic acid pathway intermediate and the pentose phosphate intermediate is shown. Enzymes and genes (in italics) involved in the bioprocess are mentioned on both sides of the arrows. PEP, phosphoenolpyruvate; E4P, erythrose 4-phosphate; DAHP, 3-deoxy-D-arabino-heptulosonate; S7P, sedoheptulose-7-phosphate; 3-DHQ, 3-dehydroquinate; EV, 2-epi-5-epi-valiolone.

Solar radiation intensity and spectrum are additional factors that influence the biosynthesis of MAAs [50]. MG and serine are condensed by an NRPS-like enzyme with the gene *ava_3855*, producing shinorine as a result. O-MT is encoded by the gene *ava_3857*, while the enzyme S7P cyclase, known as 2-epi-5-epivaliolone synthase (EVS), is encoded by the gene *ava_3858*. The enzymes along with their genes that are involved in biosynthesis of MAAs in different organisms are listed in Table 1. Together, these gene products catalyze the production of 4-DG, the original precursor of mycosporine [51]. Demethyl-4-deoxygadusol synthase and O-MT enzyme are synthesized by genes such as *ava_3858* and *ava_3857*, respectively, to produce 4-DG. The product *ava_3856* promotes the addition of glycine to 4-DG to produce mycosporine-glycine (MG) [31]. The 4-DG acts as the prominent precursor utilized in both pathways to produce all MAAs [52].

**Table 1.** The enzymes along with their genes that are involved in the biosynthesis of MAAs in different organisms.

| Organism | Genes (Protein ID) | | | | | Accession No. |
|---|---|---|---|---|---|---|
| | **DDG Synthase** | **O-MT** | **ATP-Grasp** | **NRPS-like** | **D-Ala D-Ala Ligase Homolog** | |
| *Anaebena variabilis* ATCC29413 | *ava_3858* (ABA23463.1) | *ava_3857* (ABA23462.1) | *ava_3856* (ABA23461.1) | *ava_3855* (ABA23460.1) | - | CP000117.1 |
| *Nostoc punctiforme* ATCC29133 | *npun_R5600* (ACC83905.1) | *npun_R5599* (ACC83904.1) | *npun_R5598* (ACC83903.1) | - | *npun_F5597* (ACC83902.1) | CP001037.1 |
| *Aspergillus nidulans* FGSC A4 | *an6403.4* (CBF69538.1) | *an6402.4* (CBF69540.1) | *an6402.4* (CBF69540.1) | - | - | BN001301.1 |
| *Actinosynnema mirum* DSM 43827 | *amir_4259* (ACU38114.1) | *amir_4258* (ACU38113.1) | *amir_4257* (ACU38112.1) | - | *amir_4256* (ACU38111.1) | CP001630.1 |

In majority of cyanobacteria, an operon usually carries the genes for a demethyl-4-deoxygadusol synthase and an O-MT, both of which are necessary for the biosynthesis of 4-DG [53]. The biosynthetic genes *npR5600*, *npR5599*, *npR5598,* and *npR5597* are found in *Nostoc punctiforme* ATCC 29133, while *ava_3858*, *ava_3857*, *ava_3856*, and *ava_3855* are found in *A. variablilis*. The exogenous supply of S7P in *Escherichia coli* resulted in heterologous expression of the genes *npR5600–npR5598,* thereby producing MG [54]. The biosynthesis of MAAs starts with the intermediate S7P of the PPP in *N. punctiforme* ATCC 29133 and *A. variabilis* ATCC 29413 [15]. The O-MT gene (*ava-3857*) deletion in *A. variabilis* ATCC 29413 revealed that both the shikimate and PPP depend on this gene product for the biosynthesis of MAAs [46]. When the production of MAAs was studied in four cyanobacteria, such as *Anabaena* sp. PCC 7120, *A. variabilis* PCC 7937, *Synechococcus* sp. PCC 6301, and *Synechocystis* sp. PCC 6803, MAAs were only produced in *A. variabilis* PCC 7937. It was reported by genome mining that two sets of the 3-dehydroquinate synthase (DHQS) genes, *YP_324358* and *YP_324879*, were present in *A. variabilis* PCC 7937, and it was revealed by genomic region analysis that *YP_324358* contains an O-MT gene, *YP_324357*, downstream to it, while the rest of the cyanobacteria lack these. Deoxygadusol, which makes up the common core of all MAAs, is synthesized in the presence of *YP_324358* and *YP_324357* genes. In *N. punctiforme* ATCC 29133, when DHQ was combined with the DHQ synthase and O-MT homologues (*npR5600* and *npR5599*, respectively) in the presence of the cofactors: nicotinamide adeninedinucliotide (NAD$^+$), S-adenosylmethionine (SAM), and Co$^{2+}$, 6-deoxygadusol (6-DG) was not synthesized. The notion that a shikimate pathway intermediate is involved in MAA biosynthesis was disproved. The 6-DG was synthesized in the presence of SAM, NAD$^+$, and Co$^{2+}$. Therefore, it can be concluded that 6-DG and glycine are converted into MG by the *A. variabilis* gene *ava_3856*, with the involvement of ATP and Mg$^{2+}$ cofactors [55]. The pathway for the production of MAA is the same in *N. punctiforme* ATCC 29133 and *A. variabilis,* and the homologous genes (*npR5598–5600*)

are involved [56]. Figure 2 represents the MAAs biosynthetic gene clusters of different microorganisms. *NpR*5598 functions as an *ava_3856* homologue in *N. punctiforme*, although its specific activity has not been identified. Homologues of *ava_3855* are not found in the genome of *N. punctiforme*, and it was also lacking in cyanobacteria which contain MAA clusters. MG was synthesized in *E. coli* by heterologous expression of *mys*A, *mys*B, and *mys*C, represented by *npR5600*, *npR5599*, and *npR5598* genes. Based on the fact that the *N. punctiforme* gene product NpF5597 has conserved homologues that are spatially associated with the MAA cluster in numerous cyanobacteria, it shows homologies to the recognized amino acid-ligating enzymes [54].

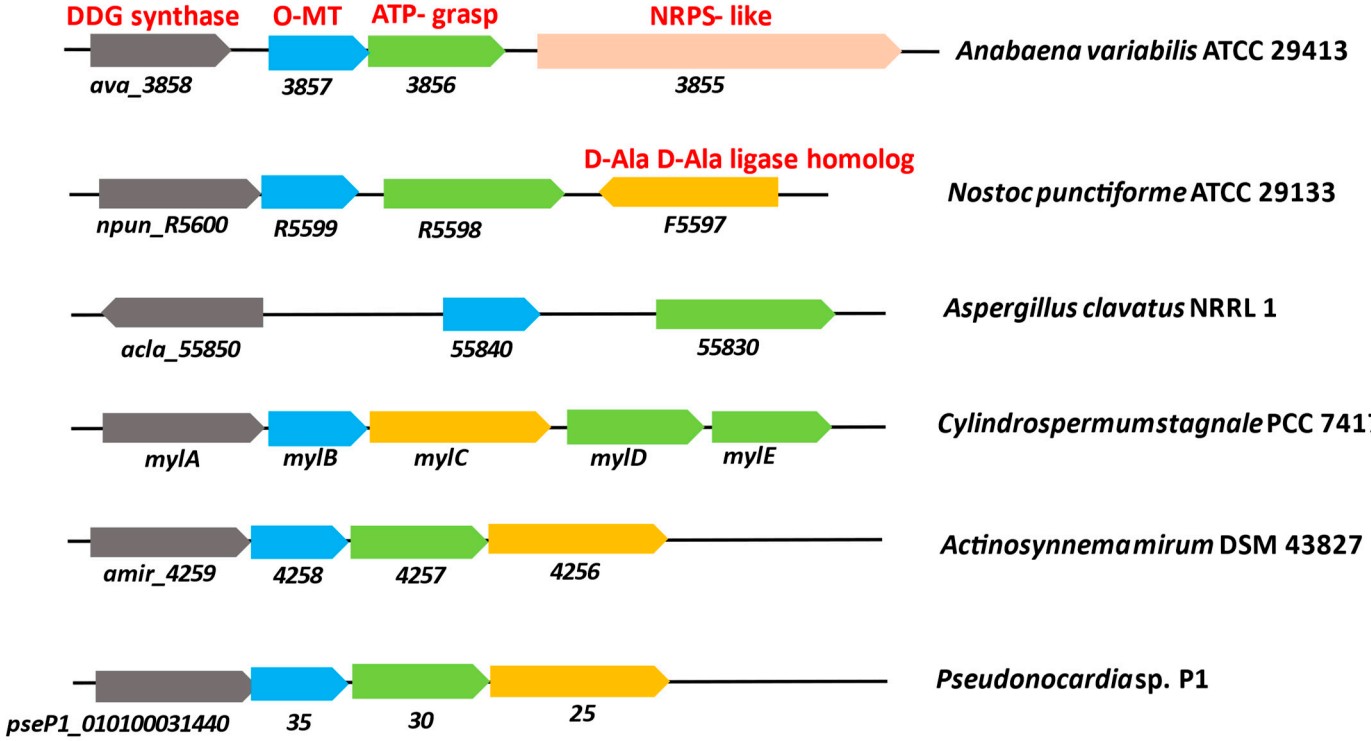

**Figure 2.** Genes involved in the biosynthesis of mycosporines and MAAs in cyanobacteria and other organisms. *A. variabilis* ATCC 29413 and *N. punctiforme* ATCC 29133 are known producers of MAAs. Comparison of genomic regions of *Anabaena variabilis* ATCC 29413 and *N. punctiforme* ATCC 29133 with *Aspergillus clavatus* NRRL 1, *Cylindrospermum stagnale* PCC 7417, *Actinosynnema mirum* DSM 43827, and *Pseudonocardia* sp. P1 (modified from Miyamoto et al. [57]).

In a genome-mining study, it was reported that production of MAAs takes place in cyanobacteria in the presence of homologs of the *EVS* gene and is absent in non-producers [58]. In *A. variabilis* ATCC 29413, the genetics of MAAs and key processes in MAAs production were studied [53].

## 4. Toolkits for Heterologous Production of MAAs

The production of MAAs from both native and heterologous hosts has been transformed by recent developments in metabolic engineering methodologies and the use of synthetic biology technologies [59]. Cloning and assembly can be the main focus for the heterologous expression of cyanobacterial MAAs, followed by BGC expression, heterologous host selection, and product optimization (Figure 3).

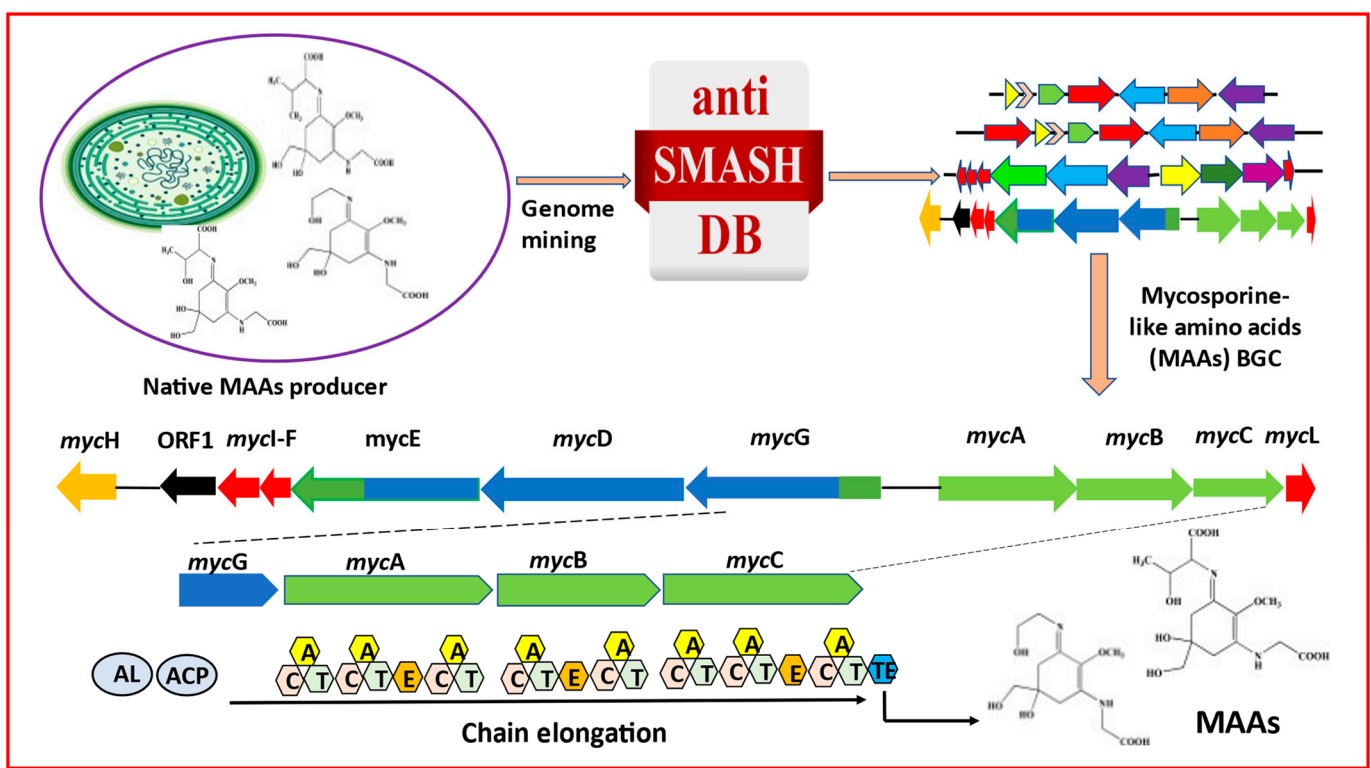

(**a**)

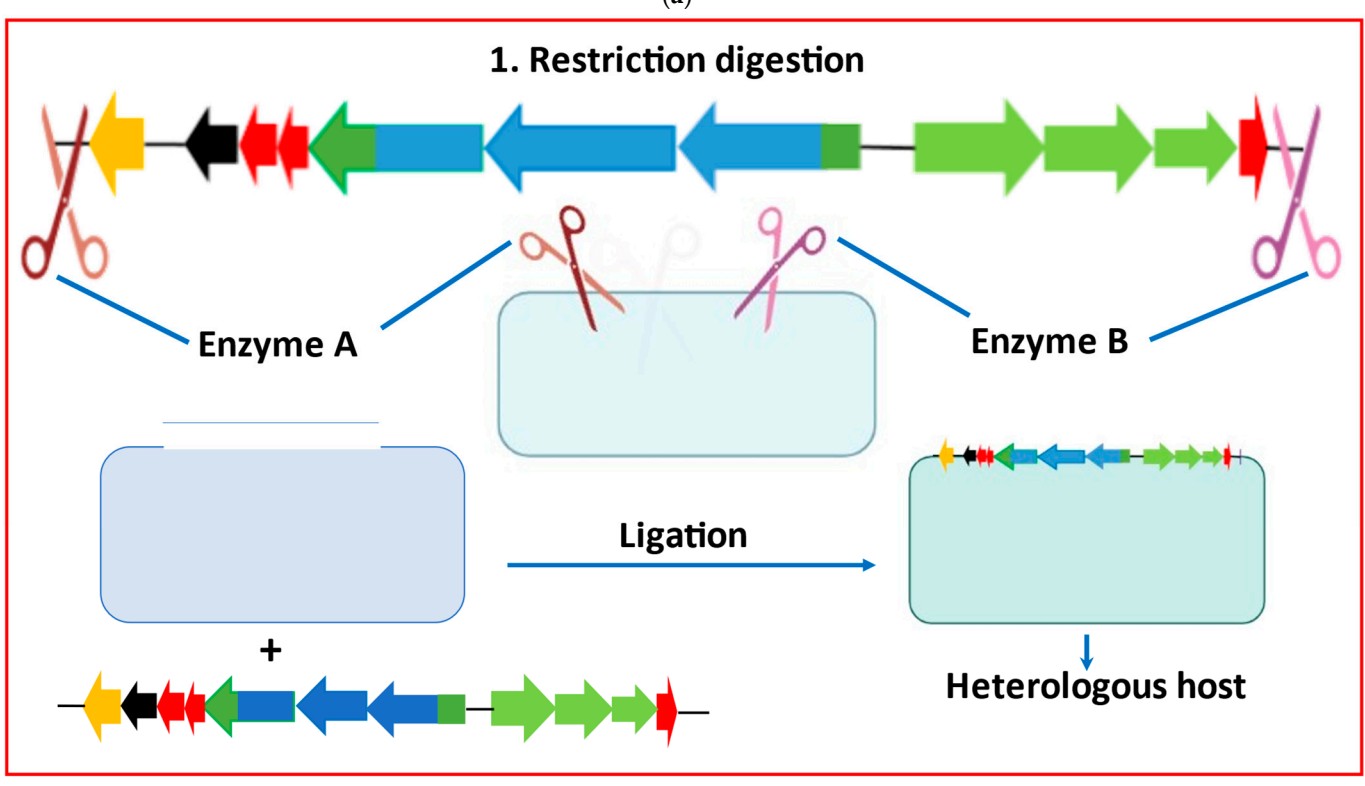

(**b**)

**Figure 3.** *Cont.*

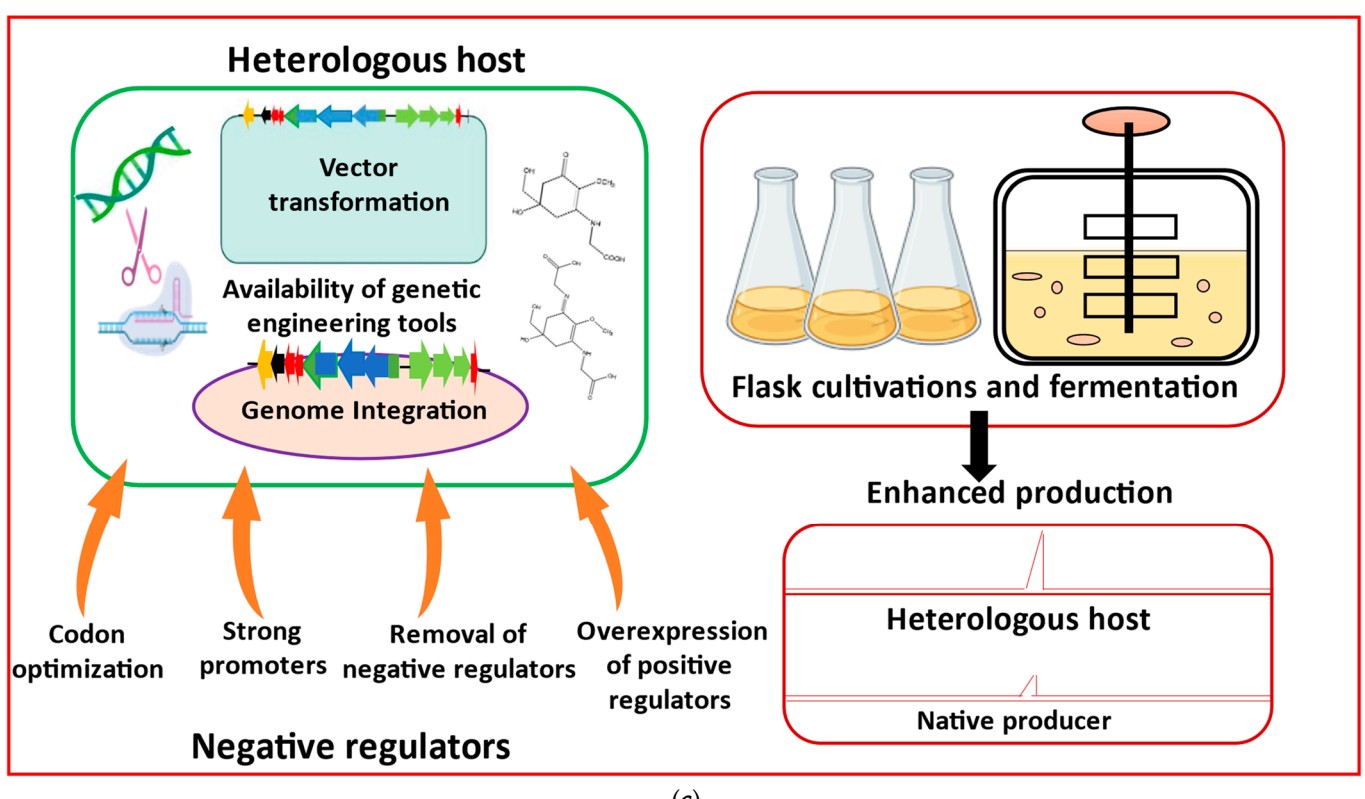

(**c**)

**Figure 3.** Major metabolic engineering and synthetic biology toolkits essential for heterologous production of mycosporine-like amino acids (MAAs) are listed as: (**a**) Genome mining using cyanobacterial genomic databases to identify BGCs responsible for MAAs biosynthesis. Domain 'A' represents adenylation, and 'C', 'T', 'E', and 'TE' represent condensation, thiolation, epimerization, and thioesterase, respectively. Acyl-CoA ligase (AL) and acyl-carrier protein (ACP) are additionally needed to initiate the secondary metabolite biosynthesis. (**b**) Restriction digestion and ligation, a primary approach of cloning and assembling BGCs, is most commonly utilized for heterologous expression of MAAs. (**c**) Considerations for heterologous expression and product analysis in heterologous hosts (modified from Sharma et al. [60]).

### 4.1. Cloning and Assembly of BGCs

Cloning, which literally means the formation of multiple copies of a genome or a gene of interest, involves fragmentation of the DNA strand to obtain the gene of interest and amplifying it, ligation of the desired sequence into the vector, transfection of the vector into the heterologous host cells, and finally, the screening or selection of the transformed host cell [61,62]. Transferring genetic materials into a heterologous host is a crucial and basic step in the heterologous production of any natural product. Cyanobacteria produce a wide variety of secondary metabolites with diverse functions, ranging from biomedical, antibacterial, antitumor, and antifungal activities [63–66]. Some of these naturally occurring cyanobacterial products are produced in small quantities, necessitating a large culture to achieve a high yield. Recently, a technique known as "biosynthetic gene clusters" (BGCs) has integrated the use of genomics and bioinformatics to address demands associated with genes [5]. For enhancing the production of MAAs in cyanobacteria, some crucial genes from the biosynthesis process of MAAs must be cloned and heterologously expressed into an appropriate host, such as *E. coli*. Biosynthetic gene clusters (BGCs) reveal the specialized gene clusters that code for secondary metabolites using the bioinformatics techniques (Figure 3a). Non-ribosomal peptide (NRP) is a type of BGC, which has diverse structures and functions and is synthesized by non-ribosomal peptide synthetases (NRPSs) [67]. The most suitable hosts for cyanobacterial BGCs are *E. coli*, *Actinobacteria*, and yeast [68]. Small BGCs are directly cloned using the conventional restriction digestion and ligation method

(Figure 3b). This method has been used to successfully clone cyanobacterial BGCs for MAAs from *C. stagnale* PCC 7417 and for shinorine from *Fischerella* sp. PCC 9339 [56,69].

As discussed above in the biosynthetic pathways, 4-deoxygadusol (4-DG), which is a common intermediate from both PPP and the shikimic acid pathway [49], is a key compound for MAAs synthesis. An enzyme from the ATP-grasp superfamily, MysC, catalyzes the conversion of 4-DG into MG by adding an amino acid, L-Gly, at the third carbon position. MG acts as an immediate precursor for the synthesis of various MAAs and contains an amino acid moiety at the C1 position. It is possible to clone these BGCs from diverse cyanobacteria into an appropriate host for the synthesis of MAAs. According to Walsh's group, the MysC enzyme from *A. variabilis* ATCC 29413, just like other enzymes of the ATP-grasp superfamily, phosphorylates 4-DG instead of L-Gly, and it was biochemically confirmed that a non-ribosomal peptide synthetase (NRPs)-like enzyme, MysE, catalyzes this step in the biosynthesis of shinorine from MG. In *N. punctiforme* ATCC 29133, BGCs lack a NRPs gene, and instead, a D-Ala-D-Ala ligase-like enzyme gene, *mys*D, is present [54]. On expression of *mys*D, BGCs in a heterologous host, such as *E. coli*, produce three MAA analogues, shinorine in majority, and porphyra-334 and mycosporine-2-Gly in minority, confirming MysD's involvement in the MAAs biosynthesis. Katoch et al. [56] isolated a gene cluster, *myl* (*myl*A–*myl*E), from *C. stagnale* PCC 741. These genes show homology to MAAs biosynthetic gene clusters of *A. variabilis* and *N. punctiforme*. The *myl* gene cluster was amplified through PCR, cloned in the pET-28b plasmid, and expressed in a heterologous host, *E. coli*, which resulted in increased production of MAAs, detected through HPLC analysis [56].

### 4.2. Heterologous Hosts for Producing Cyanobacterial MAAs

Successful heterologous production of cyanobacterial MAAs critically requires the selection of the most suitable production chassis. The high-throughput library expression of cyanobacterial biosynthetic gene clusters (BGCs) requires translational expression, availability of biosynthetic precursors, and cofactors that are essential to complete all biosynthetic steps. A suitable host comprises of certain attributes, such as a faster growth rate, easy genetic manipulations, high stability, and should be easily manageable under laboratory conditions. While many BGCs that encode cyanobacterial MAAs are known, the slow growth and lack of genetic tools in the native producers hampers their modification, characterization, and large-scale production. To produce MAAs in large quantities for commercialization, heterologous hosts can be engineered to express cyanobacterial BGCs. Numerous efforts have been undertaken to choose and enhance the most effective heterologous hosts for the efficient expression and mass manufacture of MAAs (Figure 3c). The successful production of MAAs using various heterologous hosts is listed in Table 2.

Due to its rapid growth, simple culture conditions, metabolic plasticity, and well-known genetic manipulation tools, *E. coli* is the most frequently used host for the heterologous expression of BGCs [70,71]. Mycosporine-ornithine/mycosporine-lysine are naturally produced by *C. stagnale* PCC 7417. These MAAs were effectively expressed in *E. coli* BL21(DE3) utilizing the restriction digestion cloning method [56].

Other cyanobacteria and actinobacteria were identified to have homologous gene clusters [72]. Among the 30 MAAs that are reported in cyanobacteria, shinorine comprising of glycine and serine constituents is generally used for the production of natural sunscreens [73]. Majority of the research has focused on using hosts such as cyanobacteria, *E. coli*, *Streptomyces*, and *Corynebacterium* to boost the supply of shinorine, a naturally occurring sunscreen [57,69,72,74]. From the filamentous cyanobacterium *Fischerella* sp. PCC9229, shinorine biosynthetic genes were expressed in the cyanobacterium *Synechocystis* sp. PCC6803, yielding 2.37 mg/g DCW (0.71 mg/L) of shinorine, as reported by Yang et al. [69]. The gene expression levels were improved by using multiple promoters.

MAA biosynthetic gene clusters from actinobacterium *Actinosynnema mirum* DSM 43827 were expressed in *Corynebacterium glutamicum*, leading to the production of 19 mg/L of shinorine, as reported by Tsuge et al. [74]. However, it was reported that via the expression

of the same biosynthetic genes from *A. mirum* DSM 43827 in *Streptomyces avermitilis* SUKA22, the yield was 13.9 mg MAAs/g DCW, which accounted for 154 mg/L of shinorine and 188 mg/L of total MAA [57].

Due to its safety and well-researched metabolic pathways, *S. cerevisiae* can be a viable host for heterologous expression of MAAs [75]. After extensive host engineering work, the shinorine BGC from *N. punctiforme* and xylose assimilation genes from *Scheffersomyces stiptis* were expressed in yeast. The introduction of xylose assimilation genes was an effort to boost the S7P pool to make it accessible for shinorine production. These genetic manipulations in yeast resulted in the production of 31.0 mg/L of shinorine, as reported by Park et al. [73].

Tsuge et al. [74] utilized gluconic acid (GA) as a carbon source for shinorine production, where GA acts as a carbon flux to the PPP and increases the production of an intermediate S7P. When the operon genes (*amir*4256, *amir*4257, *amir*4258, and *amir*4259) responsible for shinorine synthesis were introduced into a Gram-positive bacterium, *Corynebacterium glutamicum*, no shinorine production was noted in the presence of the lactate dehydrogenase (ldhA) enzyme. In an ldhA-deficient strain (named YTK674), an unquantifiable amount of shinorine was produced because it inhibited the lactic acid production in the late exponential and stationary phases, which prevented the pH decrease and thus prolonged the cell growth. Deletion of the *tal* gene in the ldhA-deficient strain increased the accumulation of S7P, leading to 5.3-fold higher shinorine production (Tsuge et al. [74]).

**Table 2.** Production of MAAs in heterologous hosts.

| Name | Origin of MAAs Gene Cluster | Heterologous Hosts | Cloning Method | Culture Conditions | MAAs | | References |
|---|---|---|---|---|---|---|---|
| | | | | | Content (mg/g DCW) | Titer (mg/L) | |
| Shinorine | *Anabaena variabilis* ATCC 29413 | *E. coli* | - | Batch, LB medium, 20 °C, 20 h, 500 mM IPTG induction | - | 0.15 | [72] |
| Shinorine | *Fischerella* sp. PCC9339 | *Synechocystis* sp. PCC6803 | RDL | Batch, BG11 medium (0.5 mM serine), 26 °C, 13 days | 2.37 | 0.71 | [69] |
| Shinorine | *Nostoc punctiforme* | *Saccharomyces cerevisiae* | RDL | Batch, SC-Trp medium (8 g/L xylose and 12 g/L glucose), 30 °C, 120 h | 9.62 | 31.0 | [73] |
| Mycosporine-ornithine/mycosporine-lysine | *C. stagnale* PCC 7417 | *E. coli* BL21 (DE3) | RDL | Batch, LB medium with 50 µg · mL$^{-1}$ kanamycin and 34 µg·mL$^{-1}$ chloramphenicol, up to 1 M IPTG induction | - | - | [56] |
| Shinorine | *Actinosynnema mirum* DSM43827 | *Streptomyces avermitilis* | RDL | Batch, synthetic production medium (60 g/L glucose and 400 mM NH4Cl), 28 °C, 8 days | 11.4 | 154.1 | [57] |
| Shinorine | *Actinosynnema mirum* DSM43827 | *Corynebacterium glutamicum* | Infusion cloning (ligation-independent) | Fed-batch, brain heart infusion (BHI) medium (40 g/L sodium D-gluconate and 0.5% CaCO$_3$), 30 °C, 72 h | - | 19.1 | [74] |
| Shinorine | *Nostoc punctiforme* | *Saccharomyces cerevisiae* JHYS133-6 strain | RDL | Batch, SC-Ura medium, overexpression of *Ava3858* gene | 17.0 | 47.7 | [76] |
| Shinorine | *Nostoc punctiforme* | *Saccharomyces cerevisiae* JHYS133-6 strain | RDL | Batch, SC-Ura medium (6 g/L xylose and 14 g/L glucose), 30 °C, 170 rpm shaking | - | 68.4 | [76] |

### 4.3. Product Optimization in Engineered Hosts

A remedy for the poor yield production in the native organisms can be found by engineering heterologous hosts to increase the synthesis of target compounds. According to the needs of the target molecules, heterologous hosts can be engineered in a variety of ways. For example, the target compound's metabolic pathway genes can be isolated, competitive pathways can be eliminated, biosynthetic pathway genes can be incorporated into suitable host vectors, the supply of suitable biosynthetic precursors and cofactors can be increased, and methods for maintaining and optimizing the target metabolic pathway can be chosen. Katoch et al. [56] reported a new gene cluster (*myl*) responsible for MAA biosynthesis in the cyanobacterium *C. stagnale* strain PCC 7417. The *myl* gene cluster was homologous to MAA gene clusters (*myl*A to *myl*E) from certain cyanobacteria, such as *N. punctiforme*, *A. variabilis,* and *Aphanothece*. Biosynthetic genes *myl*A to *myl*E were cloned and expressed in *E. coli*. Analytical-scale HPLC was used to determine the compounds encoded by these genes. The *myl* gene cluster expressed in *E. coli* is responsible for the production of mycosporine-ornithine and mycosporine-lysine.

The heterologous expression of the cyanobacterial-targeted metabolite, shinorine, in *S. cerevisiae* is an excellent example of host engineering [73] (Figure 4). The shinorine BGC genes from *N. punctiforme* encoding DDGS (*npR5600*), O-MT (*npR5599*), ATP-grasp ligase (*npR5598*), and D-ala-D-ala ligase (*npR5597*) were cloned into a multigene expression vector, and the multiple copies were integrated at the Ty retrotransposon delta sites in the yeast by homologous recombination [74,77]. Cloning was carried out under the control of P$_{TDH3}$ or P$_{TEF1}$, constitutive promoters resulting in generation of the coex413-NpR4 plasmid. The control strain comprising of the p413GPD plasmid did not show any production of shinorine, while *S. cerevisiae* comprising of the coex413-NpR4 plasmid showed production of shinorine (0.46 mg/L and 0.085 mg/g DCW), suggesting the presence of functional biosynthetic genes for the production of shinorine in *S. cerevisiae* [73].

Even though numerous biosynthetic genes' integration has shown enhanced shinorine production, the levels were still quite low. Low productivity may be caused by a shortage of S7P, the substrate for the first enzyme, DDGS, in the shinorine biosynthesis pathway. Due to the fact that S7P is an intermediary of the PPP, increased carbon flux in that direction may be a solution to S7P shortage [68]. Three xylose assimilation genes, *xyl*1 encoding xylose reductase, *xyl*2 xylose dehydrogenase, and *xyl*3 encoding xylose kinase from *S. stipites*, were introduced into the yeast. These assimilatory genes mainly function to convert xylose into xylulose-5-phosphate (X5P). X5P enters the PPP, resulting in increased levels of S7P, which is involved in the shinorine biosynthetic pathway [73]. The CRISPR-Cas9 approach was employed for overexpression of STB5 and TKL1. STB5 activates the expression of multiple genes encoding enzymes that are involved in the regulation of PPP, while TKL1 is a transketolase which combines PPP with the glycolysis by catalyzing the reactions between S7P and G3P. TAL1, a transaldolase, is involved in catalyzing the interconversion of S7P and G3P in PPP. Elimination of TAL1 mediated by CRISPR-Cas9 resulted in an enhanced cellular pool of S7P [73]. When cells were grown in medium containing 12 g/L of glucose and 8 g/L of xylose, the highest level of shinorine production, 31.0 mg/L and 9.6 mg/g DCW, in the final engineered yeast was observed, as reported by Park et al. [73]. This describes the potential of a suitable engineered host for the heterologous production of cyanobacterial MAAs.

The research study performed by Park et al. [73] was further carried out by Jin et al. [76] to efficiently produce shinorine in the heterologous host *S. cerevisiae.* The yeast strain engineered by Park et al. [73] had a limited capacity to utilize xylose, demanding the optimal ratios of xylose and glucose for mass production of shinorine. Jin et al. [76] further attenuated the glycolytic pathway by eliminating two target genes, *hxk2* and *gcr2*, thereby driving the carbon flux towards the shinorine production. Hxk2, a hexokinase, catalyzes the conversion of glucose into glucose-6-phosphate. Hxk2 is the main enzyme involved in glucose utilization [78]. The transcription factor Gcr1 is essential in activating glycolytic genes. Gcr2 acts as an activator molecule for Gcr1 [79]. Hxk2 is involved in regulation of

glucose repression. Deletion of Hxk2 results in glucose de-repression, favoring shinorine production. Jin et al. [76] additionally expressed the gene *ava3858* from *A. variabilis* and *amir4259* from *A. mirum,* encoding for the DDGS enzyme. Overexpression of *ava3858* resulted in 2.2-fold increased production of shinorine, in comparison to that of the previous strain. However, overexpression of *amir4259* exerted a slightly negative effect on shinorine production. The highest shinorine production titer of 68.4 mg/L was reported by Jin et al. [76].

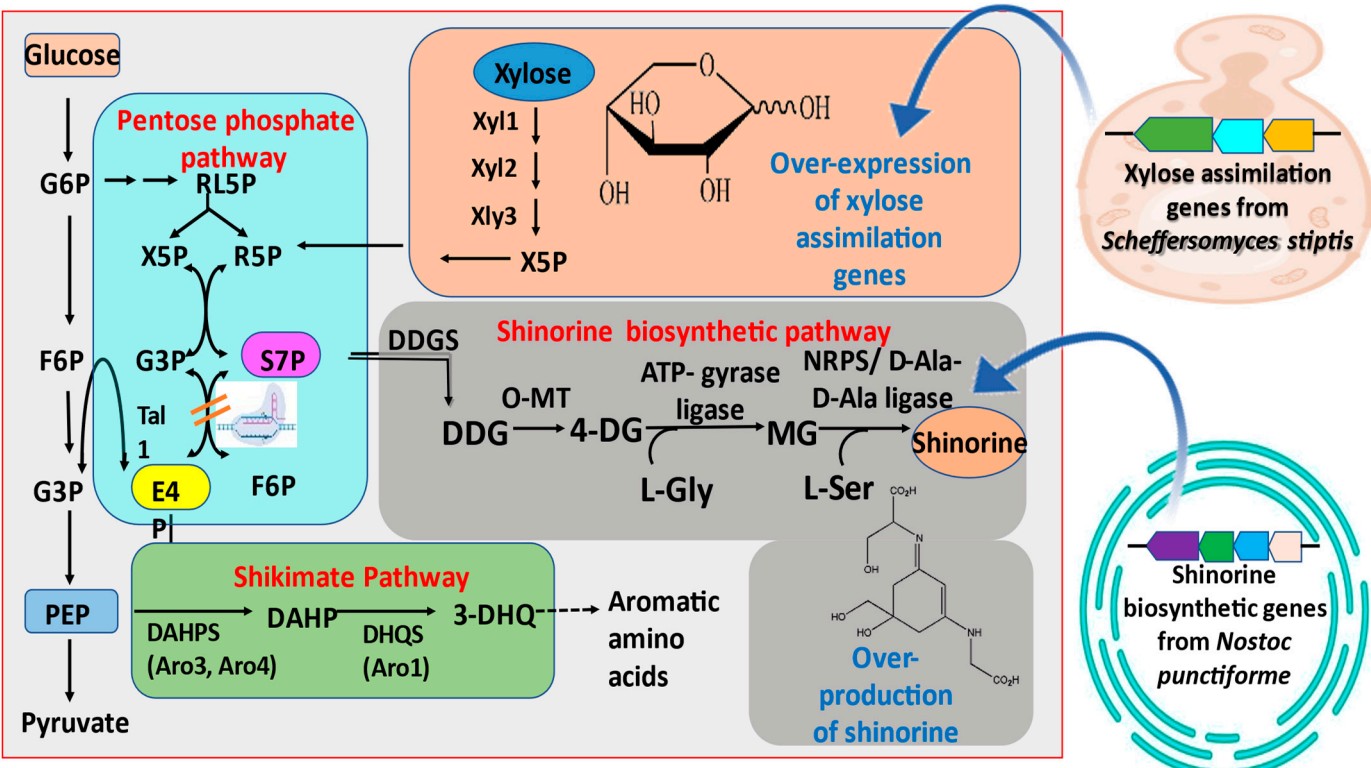

**Figure 4.** Metabolic pathway for heterologous expression of shinorine in *Saccharomyces cerevisiae.* Xylose assimilation genes from *Scheffersomyces stipitis* and shinorine biosynthetic genes from *Nostoc punctiforme* were introduced in the yeast cell. S7P is converted to shinorine via DDG, 4-DG, and MG by sequential catalytic reactions of DDG synthase (DDGS), O-methyl transferase (OMT), ATP-grasp ligase, and non-ribosomal peptides synthetase (NRPS), or D-ala-D-ala ligase. Xylose utilization genes encoding xylose reductase (Xyl1), xylitol dehydrogenase (Xyl2), and xylulokinase (Xyl3) increased the cellular pool of S7P, thereby resulting in increased shinorine production. G6P, glucose-6-phosphate; F6P, fructose-6-phosphate; G3P, glycerol-3-phosphate; 6PG, 6-phosphogluconate; Ru5P, ribulose-5-phosphate; R5P, ribose-5-phosphate; X5P, xylulose-5-phosphate; S7P, sedoheptulose-7-phosphate; E4P, erythrose-4-phosphate; DDG, 2-demethyl-4-deoxygadusol; 4-DG, 4-deoxygadusol; MG, mycosporine-glycine (modified from Park et al. [73]).

The recent development of effective genome engineering tools such as CRISPR-Cas has made it easier to improve the productivity of certain metabolites, including those from cyanobacterial BGCs [80,81]. The CRISPR-Cas9 technique was primarily developed to facilitate transcriptional regulation, genomic engineering, and marker-less gene deletion. Prior reviews of the engineering of *E. coli*, yeast, and *Streptomyces* strains using CRISPR-Cas systems have been published [82–84]. As previously indicated, the heterologous synthesis of shinorine in yeast has already utilized CRISPR-Cas9-based gene inactivation [73].

### 4.4. Application of Heterologously Produced MAAs in Transcriptional Modulation of Genes

Specifically, after introducing the cyanobacterial 4-gene *mys* cluster, the primary MAA, shinorine, was successfully synthesized in vitro in *E. coli*. A five-gene cluster, *myl*A–E, gener-

ated from the cyanobacterium *C. stagnale*, was used to heterologously express the MAAs in *E. coli*, which produced both mycosporine-lysine and the recently found MAA mycosporine-ornithine. Further, 4-DG, a MAA precursor, and five distinct MAAs, including shinorine, porphyra-334, mycosporine-glycine, palythine, and mycosporine-glycine-alanine, were produced as a result of the MAA gene cluster discovered in *Nostoc linckia* [52,56,85]. While few findings have been reported in *Streptomyces avermitilis* SUKA22, heterologous expression of these biosynthetic gene clusters led to the accumulation of MAAs such as porphyra-334 and shinorine [57].

Recent studies disclose how shinorine, mycosporine-glycine, and porphyra-334 modulate the transcription of various species' immune-regulatory and anti-inflammatory genes [36,86,87]. According to Suh et al. [36], mycosporine-glycine treatment significantly reduced the levels of COX-2 mRNA that are activated by ultraviolet radiation (UVR) and cause inflammation in the human keratinocyte cell line HaCaT. The expression of UVR-suppressed, elastin genes, and the procollagen C proteinase enhancer was elevated by all MAA treatments. Becker et al. [86] reported that shinorine and porphyra-334 both enhanced the activity of nuclear factor NF-κB in the NF-κB/AP-1 reporter myelomonocyte cell line THP-1-blue, but NF-κB induction was stronger with shinorine. However, porphyra-334 dramatically decreased the NF-κB response in cells activated by LPS, whereas shinorine only slightly increased the NF-κB activity that was superinduced by LPS. Porphyra-334 was previously shown to have an inhibitory impact on the production of NF-κB-dependent inflammatory genes, namely IL-6 and TNF-κB, in UV-A-irradiated skin fibroblasts, as reported by Ryu et al. [87]. Similar to that, porphyra-334 has been found by Gacesa et al. [88] to activate the nuclear factor erythroid 2-related factor 2 protein (Nrf2) signaling pathway in UV-A-exposed cells. However, there has not been any proof of nuclear Nrf2 translocation by porphyra-334 without concomitant ROS generation by UV-A radiation. They discovered that after treatment of MAAs in cells pre-exposed to UV-A-generated oxidative stress, enhanced expression of Nrf2-targeted downstream genes were found. Surprisingly, the expression levels of the glutamate-cysteine ligase modifier subunit (*gclm*), with glutamate-cysteine ligase (*gclc*) and heme oxygenase-1 (*hmox-1*) genes, were unaffected by MAA treatment in cells that had not previously experienced oxidative stress. This surprising finding enlightened that MAAs become antagonists of Kelch-like ECH-associated protein 1–nuclear factor erythroid 2-related factor 2 protein (Keap1–Nrf2) binding only after being exposed to oxidatively stressed cellular circumstances. In their investigation, Gacesa et al. [88] speculated that UV-A-activated kinases may impair the connection between Keap1 and Nrf2, allowing porphyra-334 and shinorine to interfere with Keap1–Nrf2 binding at doses inadequate to do so, in cells not exposed to UV-A. The expression of the matrix metalloproteinase-1 (*mmp-1*) gene is a well-acknowledged indicator of the photoaging caused by oxidative stress in human skin [89]. MMP-1 expression significantly dropped in cells treated with MAAs after UV-A exposure, evidencing a definite protective effect. Again, only cells that had already been exposed to radiation showed evidence of transcriptional control by MAAs. These findings appear to confirm that MAAs may activate the cytoprotective Keap1–Nrf2 pathway by competing with the Keap1 receptor. Although we hypothesize that increased MAA levels to UVR-produced oxidative stress might be beneficial for future therapeutic development of these natural products, the molecular stress-signaling pathway of this activation is obviously still unclear and requires further investigation.

## 5. Conclusions and Future Perspectives

Cyanobacteria produce a plethora of natural products, having diverse structures and biological activities. MAAs, one such natural product, are photoprotective compounds found in cyanobacteria, harboring a wide range of bioactive activities, including pharmaceuticals, biomedical, anti-cancerous, and antibacterial properties. MAAs can also be utilized to replace various synthetic chemicals used in the UV-protection sunscreens and are thus a great resource for the cosmetics industry. The major bottleneck in commercializing this

photoproduct is its low production rate and capital demand of biorefineries for larger-scale production. Recently, synthetic biology has opened the doors for large-scale production of these photoprotective compounds by utilizing the genomics and metabolomics. In our review, we discussed the approach of synthetic biology, which involves the heterologous expression of the BGCs of MAAs biosynthetic pathways into the desirable hosts. The selection of the desired BGCs, integrating them into a suitable vector, and the heterologous expression into the suitable host are a few approaches discussed in this review. Currently, researchers are becoming increasingly interested in the production of cyanobacterial natural products via heterologous expression.

The heterologous expression of MAAs can be achieved using recombinant DNA technology or metabolic engineering. Metabolic engineering may involve deletion of unwanted BGCs, including transcriptional terminators, modification of regulators, and engineering of tRNAs. The heterologous hosts (e.g., *E. coli*, yeast, and *Streptomyces*) could serve as platforms for structural diversification through precursor-directed biosynthesis, mutasynthesis, and combinatorial biosynthesis. For easily detecting and purifying heterologous MAAs production, BGCs of unwanted metabolites should be deleted from the host strains. Extensive modification of heterologous genes and the corresponding enzymes at different levels may result in better production rates. Systems and synthetic biology can be used as an integrative approach, to assist the engineering of desirable heterologous hosts for MAAs production. However, there has been relatively little research on this specific strategy of enhancing MAAs production via heterologous expression, and this needs to be further investigated. Therefore, this is a hot topic of research for the upcoming scientists.

**Author Contributions:** Study concept and design, V.K.S. and R.P.S.; manuscript writing, V.K.S., S.J., P.R., A.G., A.P.S. and N.K.; manuscript editing, V.K.S. and R.P.S.; manuscript review, V.K.S., S.M., P.R.S. and J.J. All authors have read and agreed to the published version of the manuscript.

**Funding:** This research did not receive any specific grant from funding agencies in the public, commercial, or not-for-profit sectors.

**Institutional Review Board Statement:** Not applicable.

**Informed Consent Statement:** Not applicable.

**Data Availability Statement:** Not applicable.

**Acknowledgments:** Varsha Singh (09/0013 (12862)/2021-EMR-1), Palak Rana (CSIR-JRF), Amit Gupta (09/013 (0912)/2019-EMR-I), Neha Kumari (09/013 (0819)/2018-EMR-I), and Prashant Singh (09/013 (0795)/2018-EMR-I) are thankful to the Council of Scientific and Industrial Research (CSIR), New Delhi, India, for the Junior Research Fellowship (JRF) and the Senior Research Fellowship (SRF). Sapana Jha (No. R/Dev./Sch./UGC Non-NET Fello./2022-23/52561) is thankful to BHU for providing the institutional fellowship. Ashish Singh (NTA Ref. No. 191620014505), Sonal Mishra (Joint CSIR-UGC JRF-2019/NTA Ref. No. 191620046790), and Jyoti Jaiswal (926/CSIR-UGC-JRF DEC, 2018) are thankful to the University Grants Commission (UGC), New Delhi, India, for the financial assistance in the form of the Senior Research Fellowship (SRF). The incentive grant received from IoE (Scheme No. 6031), Banaras Hindu University, Varanasi, India, to Rajeshwar Sinha is highly acknowledged.

**Conflicts of Interest:** The authors declare no conflict of interest.

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
