# Peer review of "Application of Synthetic Biology Approaches to High-Yield Production of Mycosporine-like Amino Acids"

_fermentation, doi:10.3390/fermentation9070669_

Round 1

Reviewer 1 Report

This manuscript reports different methodologies to enhance mycosporine-like amino acids (MAAs) production in cyanobacteria. The biosynthetic pathways are well explained and their biotechnological possibilities are thoroughly considered. Potential applications and presently commercial formulations including MAAs are also shown.

Some issues are to be addressed as follows:

Lines 119-123: please rewrite this paragraph.

Lines 146, 148, and 149-150: were instead of was, and contains

Table 1: "Detected by NMR or mass spectrometry" is not compatible with the heading "yield"

Line 337-338: cluster

Lines 471-474: sentence too long

MAAs' applications: could you provide information on the dose/concentrations that are effective in affording photoprotection when used as sunscreens? What is the MAAs' content used in the commercial formulations you mention?

Author Response

Reviewer 1:

Comments and Suggestions

Issue 1: Lines 119-123: please rewrite this paragraph.

Response: We thank the reviewer for taking the time to assess our MS. We’ve revised the text to address your concerns and hope that it is now clear. Please see page 6 of the revised MS, lines 208-210.

Issue 2: Lines 146, 148, and 149-150: were instead of was, and contains

Response: We thank the reviewer for pointing this out. We’ve changed [was] to [were] and [contain] to [contains] (page 6, lines 236, 237).

Issue 3: Table 1: "Detected by NMR or mass spectrometry" is not compatible with the heading "yield"

Response: We thank the reviewer for pointing this out. This observation is correct. We’ve revised the heading. Please see page 11 of the revised MS, table 1.

Issue 4: Line 337-338: cluster

Response: We thank the reviewer for taking the time to assess our MS to point this out. We’ve changed [custer] to [cluster] (page 13, line no. 437).

Issue 5: Lines 471-474: sentence too long       

Response: We thank the reviewer for pointing this out. We’ve revised the text to address your concerns and hope that it is now clear. Please see page 2 of the revised MS, lines 87-89.

Issue 6: MAAs' applications: could you provide information on the dose/concentrations that are effective in affording photoprotection when used as sunscreens? What is the MAAs' content used in the commercial formulations you mention?

Response: We appreciate the reviewer’s insightful suggestions and agree that further elaborating on the point “dose/concentrations that are effective in affording photoprotection when used as sunscreen” would be helpful. We’ve revised the text to address your concerns and have modified the content of MAAs’ applications as per your suggestions. Please see page 3 of the revised MS, lines 106-133.

Reviewer 2 Report

The review is well-written, and can be accepted in the present form.

Author Response

Reviewer 2:

Comments and Suggestions for Authors

Issue 1: The review is well-written, and can be accepted in the present form.

Response: We thank the reviewer for his kind words and taking the time to assess our MS.

Reviewer 3 Report

This manuscript by Singh et al. tried to deal with application of synthetic biology approaches to high yield production of mycosporine-like amino acids. However, the information gathered by authors is very poor. In the most of the section, authors just discussed about the general approach of synthetic biology. Very few information was gathered regarding to application of synthetic biology in MAAs production. In addition, there are many unbalanced sentences, grammatical errors and typos giving a bumpy flow to the reading. Thus, the reviewer thought this paper need thorough revision and should not consider for the publication in current status.

There are many unbalanced sentences, grammatical errors and typos giving a bumpy flow to the reading. 

Author Response

Reviewer 3:

Comments and Suggestions

Issue 1: This manuscript by Singh et al. tried to deal with application of synthetic biology approaches to high yield production of mycosporine-like amino acids. However, the information gathered by authors is very poor. In the most of the section, authors just discussed about the general approach of synthetic biology. Very few information was gathered regarding to application of synthetic biology in MAAs production. In addition, there are many unbalanced sentences, grammatical errors and typos giving a bumpy flow to the reading. Thus, the reviewer thought this paper need thorough revision and should not consider for the publication in current status.

Response: We thank the reviewer for taking the time to assess our MS. We apologize that the  information gathered is not up to the mark. However, there has been relatively little research on this specific strategy of enhancing MAAs production via heterologous expression, and much more has to be investigated. Moreover, we have tried our best to gather the information regarding much of the work that has been done in this field till date. Additionally, we’ve corrected the typos and grammatical errors to the best of our ability in the revised MS.

Issue 2: There are many unbalanced sentences, grammatical errors and typos giving a bumpy flow to the reading. 

Response: We thank the reviewer for pointing this out. We’ve corrected the typos and grammatical errors to the best of our ability in the revised MS.

Round 2

Reviewer 3 Report

Major:                                                  

Comment 1:  Authors are also suggested to provide the table summarizing the MAAs biosynthesis pathways found in different organism in terms of enzyme name, if any homologous present, gene name, gene ID.

Comment 2: DAHP is synthesized through shikimate pathway using phosphoenolpyruvate and erythrose-4-phoshphate (E4P). E4P is an intermediate of non-oxidative phase of pentose phosphate (PP) pathway. According to Fig 1, only sedoheptulose-7-phosphate is synthesized through PP pathway. Authors are suggested to remake the figure 1 showing the precursors needed for synthesis of DAHP.

Comment 3: In the section 2, Line 174-179, authors writing is very strange. Authors stated shikimate and PP pathway are two different MAAs biosynthetic pathways. Among these two-pathway shikimate pathway is major. Authors are strongly recommended to clarify it. Actually, shikimate pathway and PP pathway are involved in biosynthesis of precursor molecules for DHQ and EV. Shikimate pathway and PP pathway are not actually MAAs biosynthesis pathway. Authors are suggested to modify it.

Comment 4: In Fig. 2, authors showed the genetic organization of genes encoding the enzymes involved in MAAs biosynthesis pathway among different organism      . It is very difficult to follow the annotation given in figure. Which genes encode what enzymes. Authors are strongly suggested to indicate the enzyme name according to color and explain it clearly.

Comment 5: In the whole section 3, authors made many unwanted general discussions regarding to common metabolic engineering approaches (Fig 3). Authors are very strongly recommended to remove this discussion and make focus on metabolic engineering approaches that has been already used to produce MAAs. In addition to this, authors are suggested to make a table summarizing, host strain, metabolic approach used, fermentation (bioreactor or shake flask), MAAs (Titer, Yield and productivity).

Comment 6: Authors are suggested to rectify the unit of yield in table 1. Authors provided the titer and named it yield.

Moderate editing is required.

Author Response

Comments and Suggestions

Issue 1: Authors are also suggested to provide the table summarizing the MAAs biosynthesis pathways found in different organism in terms of enzyme name, if any homologous present, gene name, gene ID.

Response: We thank the reviewer for taking the time to assess our MS. We appreciate the reviewer’s insightful suggestion. We have provided the table summarizing the genes involved in MAAs biosynthesis in different organisms. Please see page 6 of the revised MS, Table 1.

Issue 2: DAHP is synthesized through shikimate pathway using phosphoenolpyruvate and erythrose-4-phoshphate (E4P). E4P is an intermediate of non-oxidative phase of pentose phosphate (PP) pathway. According to Fig 1, only sedoheptulose-7-phosphate is synthesized through PP pathway. Authors are suggested to remake the figure 1 showing the precursors needed for synthesis of DAHP.

Response: We apologize for the inappropriate information conveyed by Fig. 1. We have modified the figure as per your suggestion and hope that it is now clear. Please see page 5 of the revised MS, Fig. 1.

Issue 3: In the section 2, Line 174-179, authors writing is very strange. Authors stated shikimate and PP pathway are two different MAAs biosynthetic pathways. Among these two-pathway shikimate pathway is major. Authors are strongly recommended to clarify it. Actually, shikimate pathway and PP pathway are involved in biosynthesis of precursor molecules for DHQ and EV. Shikimate pathway and PP pathway are not actually MAAs biosynthesis pathway. Authors are suggested to modify it.

Response: We thank the reviewer for pointing this out. We’ve revised the text to address your concerns and hope that that information conveyed is now clear. Please see page 4 of the revised MS, lines 174-183.

Issue 4: In Fig. 2, authors showed the genetic organization of genes encoding the enzymes involved in MAAs biosynthesis pathway among different organism. It is very difficult to follow the annotation given in figure. Which genes encode what enzymes. Authors are strongly suggested to indicate the enzyme name according to color and explain it clearly.

Response: We thank the reviewer for pointing this out. This observation is correct. We’ve modified the figure and hope that it is now clear. Please see page 8 of the revised MS, Fig. 2.

Issue 5: In the whole section 3, authors made many unwanted general discussions regarding to common metabolic engineering approaches (Fig 3). Authors are very strongly recommended to remove this discussion and make focus on metabolic engineering approaches that has been already used to produce MAAs. In addition to this, authors are suggested to make a table summarizing, host strain, metabolic approach used, fermentation (bioreactor or shake flask), MAAs (Titer, Yield and productivity).

Response: We appreciate the reviewer’s insightful suggestions. We have removed the general discussions regarding common metabolic engineering approaches and focussed mainly on the works that are related to heterologous production of MAAs. Also, we have modified Fig. 3 as per suggestions. We have provided a table summarizing host strain, cloning methods, culture conditions and MAAs productivity. Please see page 12 of the revised MS, Table 2.

Issue 6: Authors are suggested to rectify the unit of yield in table 1. Authors provided the titer and named it yield.

Response: We thank the reviewer for pointing this out. This observation is correct. We’ve modified the table and hope that information conveyed is now clear. Please see page 12 of the revised MS, Table 2.

Comments on the Quality of English Language

Issue: Moderate editing is required.

Response: We thank the reviewer for pointing this out. We’ve corrected the grammatical errors to the best of our ability in the revised MS.

Round 3

Reviewer 3 Report

Authors addressed all the raised question by reviewer. The paper is now acceptable for the publication.

Minor editing is needed.